



# Isotopic composition of daily precipitation along southern foothills of the Himalayas: impact of marine and continental sources of atmospheric moisture

Ghulam Jeelani[1], Rajendrakumar D. Deshpande[2], Michal Galkowski[3], Kazimierz Rozanski[3]

[1]Department of Earth Sciences, University of Kashmir, Srinagar 190006, India
[2]Geosciences Division, Physical Research Laboratory (PRL), Navrangpura, Ahmedabad 380009, India
[3]Faculty of Physics and Applied Computer Science, AGH University of Science and Technology, Krakow 30-059, Poland

*Correspondence to*: Ghulam Jeelani (geojeelani@gmail.com)

**Abstract.** The flow of the Himalayan rivers, a key source of fresh water to more than a billion people primarily depends upon the strength, behaviour and duration of Indian Summer Monsoon (ISM) and Western Disturbances (WD), two contrasting circulation regimes of the regional atmosphere. Analysis of $^2$H and $^{18}$O isotope composition of daily precipitation collected along the southern foothills of the Himalayas combined with extensive backward trajectory modelling was used to gain deeper insight into the mechanisms controlling isotopic composition of precipitation and the origin of atmospheric moisture and precipitation during ISM and WD periods. Daily precipitation samples have been collected during the period September 2008 - December 2011 at six stations, extending from Srinagar in the west (Kashmir state) to Dibrugarh in the east (Assam state). In total, 548 daily precipitation samples have been collected and analysed for their stable isotope composition. It is suggested that gradual reduction in $^2$H and $^{18}$O content of precipitation in the studied region, progressing from $\delta^{18}$O values close to zero down to ca. -10 ‰ in the course of ISM evolution, stems from regional, large-scale recycling of moisture-driven monsoonal circulation. Superimposed on this general trend are short-term fluctuations of the isotopic composition of rainfall, having their roots in the local effects such as enhanced convective activity and associated higher degree of rainout of moist air masses (local amount effect), cyclonic storms or impact of isotopically heavy moisture of continental origin. Seasonal footprint maps constructed for three stations representing western, central and eastern portion of the Himalayan region indicate that influence of monsoonal circulation reaches western edges of the Himalayan region. While characteristic imprint of monsoon air masses (increase of monthly rainfall amount) can be completely absent in the western Himalaya, the onset of ISM period in this region is still clearly visible in the isotopic composition of daily precipitation. Characteristic feature of daily precipitation collected during WD period is a gradual increase of $^2$H and $^{18}$O content, reaching positive $\delta^2$H and $\delta^{18}$O values towards the end of this period. This trend can be explained by growing importance of moisture of continental origin as a source of daily precipitation. High d-excess values of daily rainfall recorded at the monitoring stations (38 cases in total, range from 20.6 to 44.0 ‰) are attributed to moisture of continental origin released into the atmosphere during evaporation of surface water bodies and/or soil water evaporation.



## 1 Introduction

The Himalayas are a 2900 km long, west-northwest to east-southeast trending mountain range, which began to form between 40 and 50 Ma ago due to the collision of two large landmasses, India and Eurasia. This immense mountain range shapes the climate of southeast Asia and the northern Hemisphere (e.g. Clemens et al., 1991; Molnar et al., 2010). The regional climate

of the Himalayas is dominated by two distinct circulation regimes of the regional atmosphere: the Indian Summer Monsoon (ISM) and the Western Disturbances (WD) periods.

The ISM is one of the most energetic components of the earth's climate system, which develops in response to the movement of Inter Tropical Convergence Zone (ITCZ) that separates wind circulation of the northern and southern hemispheres (e.g. Gadgil, 2003; Allen and Armstrong 2012). The warming of the Tibetan Plateau relative to the Indian

Ocean, resulting in low pressure over Asia and higher pressure over the Indian Ocean (Overpeck et al., 1996), pulls moisture from Southeast Asia and the Bay of Bengal and transports it north-westward (Hren, et al., 2009; Li et al., 2016). There is an east-west gradient in monsoonal influence across the Himalaya with the central Himalayas receiving according to some estimates up to 80% of its annual precipitation during the monsoon months (Bookhagen and Burbank, 2010; Lang and Barros, 2004).

Western Disturbances are eastward moving synoptic low-pressure systems embedded in the lower to mid-tropospheric westerlies in subtropics and originate from the Mediterranean Sea or mid-West Atlantic Ocean (Dimri et al., 2004; Dimri et al. 2015; Maharana and Dimri, 2014; Madhura et al., 2015, Rao and Srinivasan, 1969; Pisharoty and Desai, 1956). At times, secondary system of winds with embedded troughs develop over the Persian Gulf and the Black Sea either directly or as a result of the arrival of low-pressure systems from southwest Arabia (Dimri et al., 2004). Western Disturbances cause heavy

precipitation (>50 % of annual precipitation in winter) in the western Himalayas (Lang and Barros, 2004) and northern India from December to April (Pisharoty and Desai, 1956; Mooley, 1957; Agnihotri and Singh, 1982). They are also found to be active during summer months as well, but with low frequency (Dhar et al., 1984; Dimri, 2006). When the troughs in the mid-tropospheric westerlies extent southwards into lower latitudes, the WD reach Afghanistan, Pakistan and India and get intensified by the moisture drawn from the Arabian Sea (Chand and Singh, 2015). Cannon et al., (2015) have shown that the

heavy precipitation events occurring in western and central Himalayas due to WD are spatiotemporally independent. The strength and frequency of WD for the last three decades (1979-2010) has increased in western and decreased in central Himalayas. On the basis of the variation in the d-excess values of Ganga river water at Rishikesh, it was suggested that the snow-melt and ice-melt component has a significant fraction derived from winter precipitation with moisture source from mid-latitude westerlies (Maurya et al., 2011). The impact of climate change on the frequency and magnitude of precipitation

events in the tropics is still a matter of debate (Held and Soden, 2006; Wentz et al., 2007; Allan and Soden, 2008). This may be partly due to inadequate understanding of the atmospheric water vapour dynamics (IPCC, 2013).

The flow of the Himalayan rivers, a key source of fresh water to more than a billion people (Ives and Messerli, 1989) primarily depends upon the strength, behaviour and duration of WD and ISM. The rivers support one of the most heavily





irrigated regions in the world (Tiwari et al., 2009) and hydropower generation, the backbone of the region's economy (Karim and Veizer, 2002; Archer et al., 2010; Jeelani et al., 2012). Abnormal precipitation brought by WD and ISM can lead to flooding or drought, thereby affecting regional economies. It is therefore important to study spatiotemporal variability of ISM and WD in the Himalayas and to better characterize its causes and consequences. This implies, among others, better

understanding of the sources of atmospheric moisture forming precipitation in the region during the two contrasting circulation regimes of the lower atmosphere.

The stable isotope composition of oxygen and hydrogen in water molecules is a powerful tool for studies of the hydrological cycle, both with respect to its present status as well as its past behaviour. In the modern environment, the isotopic composition of precipitation serves as a conservative tracer for the origin, phase transitions, and transport pathways

of water (e.g. Dansgaard, 1964; Rozanski et al., 1993; Gat, 1996; Araguas-Araguas et al., 2000). The $^{18}$O and $^{2}$H content in precipitation are controlled by a) atmospheric parameters such as temperature, degree of rainout of the moist air masses, and the amount of rainfall (e.g. Dansgaard, 1964; Yurtsever and Gat, 1981; Rozanski et al., 1982; Rozanski et al., 1993), and by b) geographic factors such as altitude, latitude, moisture sources and atmospheric transport processes (e.g. Craig, 1961; Siegenthaler and Oeschger, 1980; Gat, 1996; Kendall and Coplen, 2001; Karim and Veizer, 2002). The $\delta^{2}$H and $\delta^{18}$O values

in precipitation are tightly correlated and form a so-called Global Meteoric Water Line (GMWL) in the $\delta^{2}$H-$\delta^{18}$O space, defined by the following relationship: $\delta^{2}$H=8·$\delta^{18}$O+10 (Craig, 1963). A secondary isotope parameter, deuterium excess (d = $\delta^{2}$H - 8·$\delta^{18}$O; Dansgaard, 1964) defines the position of data points in the $\delta^{2}$H-$\delta^{18}$O space with respect to GMWL. The isotopic patterns of precipitation in the tropics are expected to be different from the sub-tropics and temperate regions due to large scale convection systems, cyclonic storms and multitude of vapour sources (e.g. Midhun et al., 2013; Lekshmy et al.,

2014; Lekshmy et al., 2015). Consequently, the well-established isotope effects such as amount effect, temperature effect and altitude effect are not clearly visible in precipitation isotope data sets available for the Indian subcontinent (Deshpande and Gupta, 2012; Deshpande et al., 2010; Warrier et al., 2010).

The present study was launched with three major objectives: (i) to assess the seasonal variability of $\delta^{18}$O and $\delta^{2}$H in daily precipitation along the southern foothills of the Himalayas, (ii) to identify dominant vapour sources for precipitation in this

region, and (iii) to demarcate the influence of Indian Summer Monsoon and Western Disturbances in the study area.

## 2 Study area

Daily precipitation samples were collected at six stations located along southern foothills of the Himalayas (Fig. 1 and Table 1). The stations cover the distance of almost 2900 km, from Srinagar in the west (Kashmir state) to Dibrugarh in the east

(Assam state). In total, 548 daily precipitation samples have been collected and analysed. The largest number of samples was available for Jorhat (242) and Srinagar (121) stations. The stations are open to Indian subcontinent from the south and shielded by the Himalayan massif from the north. The elevation of the stations ranges from 99 m a.s.l. (Jorhat) to 1872 m



a.s.l. (Ranichauri). The mean annual temperature varies from 13.6 $^{\circ}$C (Srinagar) to 24.2 $^{\circ}$C (Jammu) whereas the mean annual precipitation ranges from 693 mm at Srinagar to 2781 mm at Dibrugarh.

Figure 2 shows long-term (1985-2014) monthly surface air temperature and precipitation data for six stations where daily precipitation samples have been collected. The data shown in Fig. 2 were grouped into two periods (Table 2): (i) Indian
Summer Monsoon, and (ii) Western Disturbances period. The duration of ISM was defined operationally on the basis of seasonal distribution of rainfall and through examination of individual backward trajectories available for each site. Note that the duration of ISM varied from 3 months (July-September) for the stations located at the western edge (Srinagar, Jammu, Palampur) to 5 months (May-September) for the stations located at the eastern edge of the transect (Jorthat, Dibrugarh). For the central part of the transect (stations Ranichauri and Kathmandu) the onset of ISM was set at the beginning of June and
the termination at the end of September. Except for Srinagar station, the onset of ISM is marked by distinct increase of monthly rainfall amount. Although for this station the rainfall imprint of ISM onset is not present it could still be defined on the basis of backward trajectory analyses of the air masses associated with daily rainfall, as well as through characteristic $^{18}$O and $^{2}$H isotope signatures of this rainfall.

Long-term mean values of surface air temperature and cumulative precipitation amount for ISM and WD periods for the
stations where daily precipitation samples for isotope analyses were collected in the framework of this study are listed in Table 2. Also, the peak-to-peak amplitude of seasonal changes of monthly air temperature is reported. This amplitude increases gradually from the eastern (Dibrugarth: 11.7$^{\circ}$C) to the western edge of the transect (Srinagar: 23.1$^{\circ}$C), indicating progressive transition from maritime to continental climate. The mean temperatures for ISM and WD periods also differ, the former being significantly higher. The average difference is approximately 8$^{\circ}$C.
The stations not only differ with respect to annual amount of rainfall (Table 1) but also with respect to cumulative rainfall amounts for ISM and WD periods, the former being generally higher (Table 2). The striking exception is Srinagar station - here the amount of rainfall during WD period is significantly higher (543 mm) than that during ISM period (150 mm). The ratio of cumulative rainfall amount during ISM and WD periods (parameter R in Table 2) varies from 3.41 for Dibrugarh to 1.95 for Ranichauri. For Srinagar, the value of this parameter is significantly lower than one (0.28).

### 3. Methods

Central Research Institute for Dryland Agriculture (CRIDA) and India Meteorological Department (IMD) collected most of the precipitation samples under the aegis of the National Program (IWIN) for isotope fingerprinting of waters of India (Deshpande and Gupta, 2008; Deshpande and Gupta, 2012). The Indian Meteorological Department, New Delhi, from its
sub-offices, provided relevant meteorological data for the stations. Deuterium and $^{18}$O isotope composition of rainfall samples was analysed using IWIN-IRMS facility at Physical Research Laboratory (PRL) Ahmadabad, following standard equilibration method in which water samples were equilibrated with $CO_2$ (or $H_2$) and the equilibrated $CO_2$ (or $H_2$) gas was analysed in Delta V Plus using isotope ratio mass spectrometry (IRMS) in continuous flow mode using Gasbench II (Maurya





et al., 2009). The analytical uncertainty of isotope analyses (one sigma) was ±1.0‰ and ±0.1‰ for $\delta^2H$ and $\delta^{18}O$, respectively.

The available isotope dataset was screened for negative values of the d-excess. Negative values of this parameter point to significant evaporation of rainfall, either below the cloud base, on the way to the ground, or later in the rain gauge before

collection of samples, or during their storage prior to isotope analysis. As all these effects lead to significant alteration of the original isotope signatures of the collected precipitation, the data with negative d-excess values were not considered in subsequent evaluation of the available isotope dataset. In total, 35 isotope records (ca. 6.3 %) were marked as locally affected. The largest number of excluded records belongs to the isotope dataset available for Jammu station (19 out of 98 collected there).

Reconstruction of backward trajectories of the air masses arriving at sampling stations was done through the framework of the Hybrid Single Particle Lagrangian Integrated Trajectory model (Hysplit4, revision February 2016 - Stein et al., 2015). The model was driven by archived Global Data Assimilation System (GDAS1 product, available at ftp://arlftp.arlhq.noaa.gov/pub/archives/gdas1) meteorological data available for every 3 h at 1.0°x1.0° horizontal resolution (corresponding to approx. 100 km x 100 km), with 23 sigma pressure layers between 1000 hPa and 20 hPa (Parrish and

Derber, 1992). All trajectories were calculated with temporal resolution of 30 minutes. For each location, the trajectory release point has been set up at 500 m above the local ground level, in order to represent mean elevation of moist air masses.

10-day backward trajectories representing daily rainfall samples were calculated as trajectory ensembles, each consisting of twenty seven ensemble members released at 12:00 LT on the day with precipitation sample collection. Ensembles were produced by varying the initial trajectory wind speeds and pressures, according to the Hysplit ensemble algorithm, in order

to account for the uncertainties involved in the simulation of individual backward trajectories. A slight modification of the default algorithm was used, with horizontal range of sampling of the initial values reduced from 1 to 0.5 width of the grid cell in the driving meteorological field.

For footprint analysis, individual 10-day backward trajectories starting from 12:00 LT were calculated for each station collecting daily rainfall and for Kathmandu station, Nepal, representing central region of the transect. Daily releases over the

course of three consecutive years (2009-2011) were simulated. The chosen period corresponds with the period of precipitation sampling at the stations. Daily trajectories calculated over the three-year period were then aggregated to produce the footprint maps. Footprints representative for ISM and WD seasons were calculated using the subsets of available trajectories. The output was a 0.5°x0.5° footprint signal, which was later smoothed spatially using focal averaging method.

**4 Results**

**4.1 Isotope characteristics of daily precipitation**

Figure 3 shows seasonal changes of $\delta^{18}O$ and deuterium excess for two stations with largest number of daily isotope data available (Jorhat - 242 data points, and Srinagar - 110 data points). These two stations are located on western (Srinagar) and eastern (Jorhat) edges of the studied transect (cf. Fig. 1). They have been selected to illustrate seasonal evolution of the





isotopic composition of daily precipitation at the stations along the transect. The ISM period (July-September for Srinagar and May-September for Jorhat), is marked in Fig. 3 by blue and red shading, respectively. Figure 4 shows daily $\delta^2H$ and $\delta^{18}O$ data for both stations, grouped into ISM and WD period and plotted on $\delta^2H$-$\delta^{18}O$ diagram. Worth to note is relatively large percentage of positive $\delta$ values in Fig.4. To better characterize daily precipitation data characterized by positive $\delta$ values, in

Fig.5 the d-excess values recorded for such events are plotted as a function of (positive) $\delta^{18}O$ values and the humidity deficit in the near-ground atmosphere (1 - RH, were RH stands for relative humidity). The data shown in Fig.5 represent all six stations along the transect.

Daily isotope data available for all six stations were also grouped into ISM and WD periods and resulting summary statistics are presented in Table 3. The onset and termination of ISM period at each station was chosen based on long-term

statistics of monthly rainfall (Table 2, Fig. 2) and examination of individual backward trajectories available for each site. Arithmetic averages of three isotope parameters ($\delta^2H$, $\delta^{18}O$, d-excess) with their respective uncertainties were calculated for both periods considered. Arithmetic averaging was chosen in order to better reflect average conditions at moisture sources and in the regional atmosphere during the considered period. The box-and-whisker plots of $\delta^{18}O$ and d-excess data available for each station and season are shown in Fig. 6. The $\delta^{18}O$ values of daily precipitation available for each station were

correlated with daily surface air temperature and rainfall amount recorded during analysed precipitation events. The correlations were calculated separately for ISM and WD periods. The last two columns of Table 3 summarize those calculations.

### 4.2 Backward trajectory analysis

Backward trajectory modelling was performed for all daily precipitation events analysed in the framework of this study (548 events). Figures 7 and 8 show examples of typical backward trajectories (ensembles) representing ISM and WD periods, reconstructed for Jorhat and Srinagar stations, representative for eastern and western edge of the transect, respectively. Lower parts of Figs. 7 and 8 show evolution of selected parameters of the air parcels transported along the trajectories: (i) elevation above the local ground (m), (ii) terrain height (m a.s.l.), (iii) velocity of the air parcel (m s$^{-1}$), (iv) temperature of

the air parcel (°C), (v) precipitation rate (mm 6h$^{-1}$) and (vi) $H_2O$ mixing ratio (g kg$^{-1}$). Great seasonal contrast in the origin and transport routes of the air masses bringing precipitation to the stations is apparent in both figures.

To better characterize the contribution of different air masses arriving at six stations collecting daily rainfall, footprint analysis was performed (cf. Sect. 3). The footprint maps were calculated for 2009-2011 period, based on daily simulations of 10-day long backward trajectories, started at each of the locations at noon local time. Footprint maps were prepared for three

stations, Jammu, Kathmandu and Jorhat, representing western, central and eastern part of the studied transect, respectively. Separate maps were prepared for ISM and WD periods. The footprint maps are presented in Fig. 9.

### 5. Discussion
### 5.1 Seasonality of isotope characteristics of daily rainfall



The $^{18}$O isotope composition of daily precipitation at Jorhat and Srinagar stations shown in Fig. 3a reveals a clear seasonality, apparently linked to temporal evolution of the circulation patterns of the regional atmosphere associated with ISM and WD periods. The seasonality in $^{18}$O content is particularly well seen in the data available for Jorhat station. During development of Indian Summer Monsoon, the $^{18}$O content in daily precipitation at this site decreases gradually, from $\delta^{18}$O

values fluctuating around 0‰ at the onset of ISM, to very negative $\delta^{18}$O values (up to ca. -15‰) recorded at its termination. During WD period the $^{18}$O content progressively increases, reaching positive δ values towards the end of this period.

Although d-excess records shown in Fig. 3b are rather noisy, it is apparent that at Srinagar site the d-excess progressively decreases, from high d-excess values recorded in January-February (d > 20‰) towards low d-excess values (d < 10‰) recorded in September. At Jorhat, temporal evolution of the d-excess values largely coincides with that observed at Srinagar.

However, during WD period, the d-excess is much more variable at this site and range from 0 to more than 40‰. As seen in Fig. 4, during the monsoon season the linear relationship between $\delta^2$H and $\delta^{18}$O is generally better defined pointing to moisture sources of similar nature and similar conditions of rainfall formation. At Srinagar site, there is a striking difference between Local Meteoric Water Lines representing ISM and WD periods. The intercept of LMWL representing ISM is more than two times lower than that representing WD period, with similar slopes of both lines. For Jorhat station these seasonal

differences in LMWL are less pronounced. On the other hand, the scatter of data points around LMWL during WD period is remarkable at this station, with d-excess values varying between zero and 40‰. Another interesting feature of the data shown in Fig. 4 is relatively large percentage of positive δ values (cf. Sect. 5.2).

As suggested by Fig. 3a, the largest heavy isotope depletion in rainfall is expected at the transition from ISM to WD period. Indeed, three most negative $\delta^{18}$O values of daily rainfall recorded during the study (Ranichauri: -19.48‰, Jorhat: -

22.79‰) were observed in September and October. On the other hand, the most positive $\delta^{18}$O values (Jammu: +5.28 and +6.70‰) were observed towards the end of WD period (April, June).

It is apparent from Table 3 and Fig. 6 that ISM and WD periods are characterized by distinct isotope signatures of precipitation collected along the foothills of the Himalayas. Average $\delta^2$H and $\delta^{18}$O values for ISM period are significantly lower than those recorded for WD period at the given station. The largest difference (ca. 6.6‰ for $\delta^{18}$O and 55 ‰ for $\delta^2$H)

was observed at Ranichauri station. Rainfall collected during ISM and WD periods differs also in mean d-excess values. During ISM the d values are generally lower than those observed during WD period. The largest difference in terms of d-excess was observed for Srinagar station (ca. 9‰ for ISM and 17‰ for WD). For other stations this difference is significantly smaller. In one case (Jammu station) this regularity is broken, where average d-excess value for ISM period is higher than that for WD period (Table 3). This, however, may stem from relatively high number of daily rainfall data with

positive $\delta^2$H and $\delta^{18}$O values at this station during WD period. There is a lack of distinct spatial trends in the mean isotope characteristics of daily rainfall along the transect during the two seasons (Fig. 6).

The $\delta^{18}$O values of daily precipitation available for each station were correlated with daily surface air temperature and rainfall amount. The correlations were calculated separately for ISM and WD periods. The last two columns of Table 3





summarize those calculations. As far as the link between $\delta^{18}O$ and local surface air temperature is concerned, positive, significant correlation was found for three stations: Srinagar (ISM period), Jammu (both periods) and Ranichauri (WD period). Significant negative correlation was observed for Jorhat and Dibrugarh stations (ISM period). On the other hand, no significant correlation of $\delta^{18}O$ with the amount of rainfall was found, except Srinagar station where $\delta^{18}O$ was negatively

correlated with rainfall amount during ISM period.

Distinct seasonality seen in the isotope characteristics of daily rainfall at the foothills of the Himalayas and discussed above should be viewed in the context of large seasonal changes in the circulation regime of the regional atmosphere, as illustrated by Figs. 7 & 8. The ISM period at Jorhat station (Fig. 7a) is dominated by low-level transport of moist air masses (water vapour content of ca. 18 g kg$^{-1}$) from the equatorial Indian Ocean and the Bay of Bengal. The calculated ensembles

have relatively low spread, reflecting large-scale, uniform movement of the moist air masses driven by monsoon circulation. The air masses reach the station from the south, without any noticeable contribution from other directions. The rainfall is generated when warm, moist air masses are lifted up over the continent and cool down from ca. 28°C to 20°C. As seen in Fig. 8a, the monsoonal air masses may reach also the Srinagar station located on the western edge of the transect. However, the ensembles shown in Fig. 8a suggest that apart from trajectories passing over the Bay of Bengal and travelling along the

southern foothills of the Himalayas in the northwest direction, a more direct route of moist monsoonal air masses (water vapour content of ca. 15 g kg$^{-1}$), starting in the Arabian Sea and crossing the Indian continent, is a more important source of rainfall for western Himalayas during this season.

During WD period, the overwhelming majority of air masses arrive at rainfall collection stations from west and northwest directions (Figs. 7b & 8b). Typically, they travel at high elevations (4-6 thousand meters above the ground), passing the

region of Black Sea and Caspian Sea, and further descend over Afghanistan and Pakistan, towards rainfall collection stations located at the western edge of the transect (Srinagar, Jammu, Palampur). These dry air masses (water vapour content around 1 g kg$^{-1}$) are picking up moisture of continental origin from relative proximity of the collection sites and their vapour content rises to approx. 3.5 g kg$^{-1}$ (Fig. 8b - lower panel). For stations located at the eastern edge of the transect (Jorhat, Dibrugarh) more western routes of air masses prevail (Fig 7b). They travel east within the latitude band of 20°N to 30°N passing

Arabian Peninsula, Persian Gulf, northern reaches of the Arabian Sea and northern India, gradually warming up and losing elevation. On their way east they gradually absorb moisture of both maritime (Arabian Sea) and continental origin. As a result of this process, their water vapour content increases from approximately 3 g kg$^{-1}$ at the eastern coast of the Arabian Peninsula to ca. 8 g kg$^{-1}$ in the proximity of the eastern edge of the transect.

Two characteristic features of $\delta^{18}O$ records shown in Fig. 3 need a more comprehensive evaluation: (i) gradual reduction

of $^{18}O$ (and $^{2}H$) content in daily rainfall during ISM period, and (ii) positive $\delta^{18}O$ and $\delta^{2}H$ values recorded from time to time not only at Srinagar and Jorhat but also at other stations of the transect, mostly during WD period.

## 5.2 Evolution of $\delta^{18}O$ during ISM period





Gradual, heavy-isotope depletion of rainfall in the course of development and recession of Indian Summer Monsoon, apparent from Fig. 3a, is not a local phenomenon restricted only to the studied transect. Such evolution of $\delta^{18}O$ in precipitation is observed over the entire region influenced by ISM. For instance, $\delta^{18}O$ of monthly rainfall at New Delhi decreases from ca. 0‰ in June to around -9‰ in September (Araguas et al., 1998; Battacharya et al., 2003). Also the stations

of IWIN programme (Deshpande and Gupta, 2008; Deshpande and Gupta, 2012) show similar effect of gradual decrease in $\delta^{18}O$ values of rainfall towards the end of ISM season in September, seen for example at Ahmedabad station in western India (Deshpande et al., 2010). The reduction of $^{18}O$ content of similar magnitude was observed for individual rainfall events collected during 2012 and 2013 monsoon period on Andaman Island, Bay of Bengal (Chakraborty et al., 2016).

The extent of heavy-isotope depletion of daily rainfall in the course of ISM evolution, observed in this study, is large (>

10‰ in $^{18}O$). Regional thermal gradients are virtually nonexistent during ISM and hence cannot be used to explain the observed heavy-isotope depletion, with sea surface temperatures in the Bay of Bengal fluctuating between 28 and 29°C (e.g. Midhun et al., 2013) and mean surface air temperatures at three low-elevation stations of the studied transect (Jammu, Jorhat, Dibrugarth) around 28°C (cf. Table 2). The gradual reduction of heavy-isotope content in precipitation is apparently a regional phenomenon inherently linked to the evolution of Indian Summer Monsoon and cannot be explained by local

effects. Local effects can be responsible for the "noise" superimposed on the seasonal $\delta^{18}O$ trend (cf. Fig. 3) but will not explain regional evolution of $\delta^{18}O$ and $\delta^2H$ values in the course of ISM. Clearly, another explanation should be considered. In this context, it is noteworthy that Deshpande et al (2015) have shown that major moisture pickup locations for precipitation at Ahmedabad gradually change from over the Arabian sea to central Indian continental areas in the later part of ISM ISM.

In the course of ISM period water availability on surface (lakes, reservoirs, streams, wetlands) and sub-surface (soil, vadose zone) environments and in ground-level atmosphere over large continental areas of India progressively increase to a substantial extent. For instance, in Assam state 9.7% of the area is under wetlands including rivers, streams and riverine wetlands. The open pan annual evaporation (2.36 mm/day) and annual potential evapotranspiration (3 mm/day) at Jorhat is lowest in the country (Rao et al., 2012). The relative humidity is generally higher during summer months from June to

November. The low values of open pan and potential evapotranspiration and high relative humidity in Assam suggest that atmosphere remains continuously loaded with locally generated moisture during summer months. As the monsoon progresses in India, enhanced soil moisture and vegetation cover lead to increased evapotranspiration and recycled precipitation. The recycling ratio, that is the ratio of recycled precipitation to total precipitation, is highest (around 25%) in northeast India, which has dense vegetation cover leading to high evapotranspiration. High precipitation recycling ratio was

found at the end of the monsoon in the month of September (Pathak et al., 2014).

Increasing amount of moisture in the lower atmosphere in the course of ISM makes the air column unstable, prone to convective activities. In should be noted here that in absence of horizontal thermal gradients between source regions of atmospheric moisture and the continent, the principal mechanism which can generate rainfall is vertical uplift and cooling of moist air masses, associated with convective systems. If horizontal and vertical extent of convective systems increase as the



monsoon progresses, it could generate the observed gradual depletion in heavy isotopes of daily rainfall in the course of ISM evolution.

Looking from a more broader perspective, a large-scale, regional recycling of atmospheric moisture can also contribute to the observed evolution of $\delta^{18}O$ and $\delta^2H$ during ISM period. Northward movement of ITCZ pulls maritime moisture from southeast Asia and the Bay of Bengal and transports it north-westward. Moist air masses are lifted by large-scale convection, loose part of their moisture content and return towards the equator as the upper branch of Hadley cell circulation. Then, they descend and mix with low-level moist air masses of oceanic origin (cf. Li et al., (2016) - Fig. 24). Descending air masses contain moisture depleted in heavy isotopes, which is then incorporated in the moist air masses transported northward, thus leading to gradual reduction of $^2H$ and $^{18}O$ content in the marine moisture constituting the main source of monsoon precipitation. Such regional recycling loop operating in the course of ISM evolution may provide the required mechanism for a gradual, large-scale reduction of $^{18}O$ and $^2H$ content in regional atmospheric moisture and precipitation during this period. Rough assessment of this effect was made assuming Rayleigh-type rainout of moist, maritime air masses (RH = 80%, T = 28°C), with the initial $\delta^{18}O$ value of moisture equal -10 ‰. It was assumed that rainout continues down to 15% of the initial water content. Then, the air masses return as an upper branch of the recycling loop and mix with the original maritime moisture. Five to six such recycling loops would be required to reduce the initial $^{18}O$ content of maritime moisture and the rainfall by approximately 10 ‰ at the end of the monsoon period, in accordance with observations.

It is likely that both mechanisms underlined above act together to produce the observed characteristic evolution of $\delta^{18}O$ and $\delta^2H$ in daily rainfall during monsoon period. The model runs of isotope GCMs available for Indian continent (e.g. Hoffman and Heimann, 1997; Midhun and Ramesh, 2016) suggest that the models tend to underestimate the amplitude of seasonal changes of $\delta^{18}O$, particularly in northern India. A more comprehensive isotope modelling of monsoon circulation would be needed to quantify the above-outlined mechanisms of moisture recycling and their impact on the measured stable isotope composition of precipitation in the region.

Finally, worth to comment are large seasonal changes in the isotopic composition of regional atmospheric moisture reservoir in response to contrasting circulation patterns of the regional atmosphere and moisture recycling mechanisms discussed above. When operation of the monsoon circulation engine is terminated in September the regional atmosphere is still loaded with moisture heavily depleted in $^2H$ and $^{18}O$. This remarkable heavy-isotope depletion of regional atmospheric moisture reservoir survives for several weeks. In fact, the most negative $\delta^{18}O$ value (-22.79‰) was measured in rainfall collected at Jorhat station on October 11, 2010. In the course of WD period, maritime moisture depleted in heavy isotopes is gradually replaced by moisture of continental origin characterized by elevated concentration of $^2H$ and $^{18}O$ (cf. Sect. ). This in turn is reflected in rising δ values of rainfall in the course of WD period.

### 5.3 Positive $\delta^{18}O$ and $\delta^2H$ values of daily rainfall



Another striking feature of the data generated in the framework of this study calling for explanation is relatively frequent appearance of positive $\delta^{18}O$ and $\delta^{2}H$ values in the isotope records available for six stations collecting daily rainfall. Positive $\delta$ values range from 0.17 to 6.70‰ for $\delta^{18}O$ and from 5.3 to 56.3‰ for $\delta^{2}H$. They constitute ca. 14% of the collected and analysed data. As seen in Fig. 5, their d-excess values decrease with increasing $\delta^{18}O$ values ($R^2$=0.125) and with increasing humidity deficit in the local atmosphere ($R^2$=0.189). Positive $\delta^{18}O$ and $\delta^{2}H$ values were recorded mostly during WD period (ca. 26% of all data available for this period, comparing to 4.5% recorded during ISM period). The only station where positive $\delta$ values were not observed during ISM period was Jorhat. The station where positive $\delta^{18}O$ and $\delta^{2}H$ values were recorded most frequently (more than 50% of the data available for WD period) was Jammu. This station is located at western edge of the transect, far away from oceanic sources of moisture.

The $^{18}O$ isotope composition of maritime moisture collected onboard a ship (mast top, ca. 25 m above sea level) cruising the Bay of Bengal during the ISM period (from July 13 till August 3, 2012) varied between ca. -10 and -14‰ (Midhun et al., 2013). If one adopts -10‰ as representative $\delta^{18}O$ value for unaltered oceanic moisture from which monsoon precipitation is formed, and assuming further that this moisture is transported towards the southern foothills of the Himalayas without any noticeable rainout effect, the expected $\delta^{18}O$ value of the first condensate would be around -1.0 ‰. It is highly unlikely that unaltered maritime moisture can reach such remote continental sites as Jammu station where positive $\delta$ values are most common. Hence, the 'first condensate' scenario cannot explain the observed positive $\delta^{18}O$ and $\delta^{2}H$ values, even if partial evaporation of raindrops on their way to the ground is considered, and another explanation should be thought of.

As the majority of positive $\delta$ values was recorded during WD period (in total 61 versus 13 cases during ISM period), the explanation should involve sources of moisture other than oceanic ones. As discussed above and shown in Figs. 7b & 8b, the air masses arriving at rainfall collection stations during WD period have their origin in the west, and contain overwhelming portion of moisture of continental origin. One can distinguish three components of backward flux of water vapour into the regional atmosphere over the continental areas, each characterized by distinct isotope signature: (i) water vapour transpired by plant cover, (ii) water vapour evaporated from bare soil, and (iii) water vapour evaporated from surface water bodies. It is a well-established fact that in the course of transpiration process, leaf-water is becoming progressively enriched in heavy stable isotopes, quickly reaching hydrologic and isotopic steady-state (e.g. Dongmann et al., 1974; Flanagan et al., 1991). Under these conditions, the isotopic composition of water vapour released into the atmosphere is isotopically identical with the source water utilized by plants. In our case the water utilized by plants originates predominantly from rainy (monsoon) season. The amount-weighted mean $\delta^{18}O$ value of ISM precipitation for three low-altitude stations (Jammu, Dibrugarh, Jorhat) is equal -6.4‰ and the d-excess is equal 12.5‰. First condensate produced from such water vapour (assumed temperature of condensation equal to +10°C) will be characterized by $\delta^{18}O$ values close to 4.3‰, which falls within the range of positive $\delta$ values of daily rainfall collected at the stations.



Apart of plant transpiration, soil water evaporation may also return to the local atmosphere water vapour, the isotopic composition of which is identical to that of the source (soil) water. However, due to much larger size of soil water reservoir when compared to leaf water, establishing steady-state isotope evaporation profile in the soil will require much longer period of time than is the case of leaf water. This is possible only under arid or semi-arid conditions, with periods between consecutive rain events long enough to establish such a steady-state profile (e.g. Zimmerman et al., 1966; Barnes et al., 1983). This is generally not the case for the study area. Thus, evaporation from bare soil in the study area will be more like evaporation from open water bodies. During such process the isotopic composition of evaporating water undergoes gradual isotope enrichment, evolving along the so-called local evaporation line in the $\delta^2$H-$\delta^{18}$O space (e.g. Gat, 1996). The mass balance considerations require that water vapour being released into the atmosphere in the course of such evaporation process is located on the local evaporation line, at the left-hand side of local meteoric water line (LMWL). This water vapour has somewhat reduced heavy isotope content when compared to water subject to evaporation and high deuterium excess (e.g. Rozanski et al., 2001).

All three processes outlined above are most probably acting together under climatic conditions characteristic for the study area and are apparently capable of delivering sufficient amounts of isotopically heavy moisture to the regional atmosphere to produce rainfall characterized by positive δ values. It is likely that generally higher and more variable d-excess of rainfall in the course of WD, sometimes reaching values higher than 40 ‰ (cf. Fig. 4), reflects varying contribution of the three sources loading moisture of continental origin into the local atmosphere during this period.

The fact that d-excess values are inversely correlated with $\delta^{18}$O values and decrease rising humidity deficit of the local atmosphere (Fig. 5), point to partial evaporation of raindrops as an additional mechanism contributing to the observed range of positive $\delta^{18}$O and $\delta^2$H values present in isotope records available for the rainfall collection stations. In particular, relatively large number of positive δ values recorded at Jammu station can partly stem from low relative humidity of the local atmosphere, characteristic for this location. However, this mechanism alone cannot fully explain the presence of positive δ values in the analysed isotope data records.

## 5.4 Significance of elevated d-excess values

Higher than the global average d-excess value of about 10‰ in meteoric waters originating from the Himalayas and Tibetan Plateau was often used to infer Mediterranean or more generally westerly derived vapour (Tian et al 2005; Hren et al., 2009; Jeelani et al., 2010; Bershaw et al., 2012). The observed high d-excess in rainfall was generally related to higher d-excess (ca. 20‰) found in the vapour generated over east Mediterranean Sea (Gat and Carmi, 1970). Here we argue that these high d-excess values do not necessarily originate from the Mediterranean Sea. In total, there were 38 data records with d > 20‰ observed during the sampling period (ca. 7.2%). Higher d-excess values occurred mostly during WD period (31 out of 38 cases). However, the highest d values were recorded during ISM period (Jammu, 34.0 and 39.1‰; Jorhat, 40.7 and 44.0‰).

The largest number of high d-excess values was recorded at Srinagar during WD period (21 out of 87 data records for this period). The only station without elevated d-excess values was Dibrugarh. Closer examination of backward trajectory



ensembles calculated for days with high d-excess values reveal that trajectories associated with daily rainfall samples characterized by high d-excess values arrive at Srinagar from northwest, west or southwest. However, as discussed above, those air masses are generally very dry and pick-up moisture of continental origin only in relative proximity of rainfall collection stations (cf. Fig. 8b). Surprisingly, high d-excess values recorded at Jammu station are almost exclusively

associated with characteristic monsoon-type circulation. In one case, recorded during WD period (31/12/2010), the air masses were circling around over the Indian subcontinent, interacting strongly with the surface. Thus, backward moisture fluxes originating from evaporation of continental surface water sources could be the source of  rainfall characterized by elevated d-excess values.

Common feature of almost all trajectories at all the precipitation sites with high d-excess is their long residence time in

relative proximity of the sampling site. This leaves plenty of time for their interaction with the surface, accommodating moisture from continental sources and releasing it in the form of rainfall. As a source of continental moisture producing water vapour characterized by high d-excess value may serve evaporation of surface water bodies (lakes, swamps, etc.) and/or non-steady-state evaporation of soil moisture. Some impact of moisture from Eastern Mediterranean with high d-excess arriving in the Himalayan region is certainly possible, although in our opinion is rather unlikely that this is an

important source of rainfall in the region. Low-level eastward moving, turbulent transport of moisture from Eastern Mediterranean towards the Himalayas will be inevitably associated with strong interaction with the surface on the way (rainfall, backward moisture fluxes) which will blur the original isotope signature of this moisture.

**5.5 Seasonality of circulation patterns and moisture sources for precipitation in the southern foothills of the**

**Himalayas**

The footprint maps shown in Fig. 9 provide a valuable insight into seasonal contrast in the circulation patterns of regional atmosphere, which in turn control the regional rainfall regime (amount, its seasonal distribution and stable isotope composition). The maps were constructed separately for ISM and WD periods, for three stations (Jorhat, Kathmandu and Jammu) representing eastern, central and western part of the studied transect, respectively.

The footprint map representing ISM period at Jorhat station clearly demonstrates overwhelming dominance of monsoon circulation bringing moisture-loaded air masses from tropical Indian Ocean and Bay of Bengal towards the eastern region of the Himalayas. There is a very small contribution (in the order of few percent) of the air masses arriving from west and northwest. Dominating influence of monsoon air masses is seen also in the central portion of the transect (Kathmandu site), although the presence of air masses originating in the Arabian Sea and crossing Indian subcontinent in the northeast direction

is also noticeable. The footprint map for Jammu station representing western Himalayas clearly shows three major types of air masses arriving at this site during ISM period: (i) maritime monsoonal air masses originating in the Bay of Bengal and travelling along southern foothills of the Himalayas, (ii) continental air masses coming from northwest direction, and (iii) the maritime air masses originating in the Arabian Sea and travelling along India-Pakistani border towards eastern Himalayas, the first being the dominating component.



During WD period, the regional circulation patterns change radically. The Jorhat station receives air masses predominantly from northern India and Pakistan, with noticeable contribution from the Bay of Bengal. The footprint map is generally more diffuse, indicating presence of continental air masses with the origin in central Asia as well as Black Sea and Caspian Sea region. Similar picture is observed for Kathmandu station, with majority of air masses coming from northern India and

Pakistan. The impact of maritime air masses (Bay of Bengal) is reduced, although still visible. The western part of the Himalaya (Jammu station) is under overwhelming influence of air masses coming from the west (Iran, Iraq, Afghanistan and Pakistan). Whereas small contribution of maritime air masses coming from eastern Arabian Sea is still visible, air masses coming from the Bay of Bengal are practically absent.

**6. Conclusions**

Isotope analyses of daily precipitation samples collected at six stations located along the southern foothills of the Himalayas allowed a deeper insight into the mechanisms controlling isotopic composition of precipitation in this important region of Indian subcontinent. Analysis of $^2$H and $^{18}$O isotope composition of daily precipitation, combined with extensive backward trajectory modelling of the air masses associated with rainfall in the studied region, allowed several important conclusions to

be drawn with respect to origin of atmospheric moisture and precipitation in two contrasting seasons (Indian Summer Monsoon and Western Disturbances).

It is suggested that gradual reduction in $^2$H and $^{18}$O content of precipitation in the region, progressing from $\delta^{18}$O values close to zero down to ca. -10 ‰ in the course of ISM evolution, stems from convective activities in the regional atmosphere and large-scale recycling of moisture driven by monsoonal circulation. Superimposed on this general trend are short-term

fluctuations of the isotopic composition of rainfall having their roots in local effects. Seasonal footprint maps constructed for three stations representing western, central and eastern portion of the Himalayan region indicate that influence of monsoonal circulation reaches the western edges of the Himalayan region. While characteristic imprint of monsoon air masses (increase of monthly rainfall amount) can be completely absent in eastern Himalaya, the onset of ISM period is still clearly visible in the isotopic composition of precipitation.

The most characteristic feature of daily precipitation collected in the region during WD period is its relatively high $^2$H and $^{18}$O content when compared to ISM period, and the presence of large number of daily rainfall samples exhibiting positive $\delta^{18}$O and $\delta^2$H values. These peculiar isotope characteristics can be explained only when dominating continental origin of moisture is postulated. Water stored in the soil during ISM period is returned to the local atmosphere during WD period through evapotranspiration processes. Backward trajectory modelling has shown that long-range transport of air masses from

the west and northwest, characteristic for WD period, occurs at high elevations and cannot bring sufficient amounts of moisture to cause precipitation in the study area. Instead, the necessary moisture is mainly of local (regional) origin stemming from transpiration of plant cover, soil water evaporation and evaporation of surface water bodies. Isotope characteristics of rainfall during WD period are consistent with this scenario. Seasonal footprint maps show that during that





period eastward moving air masses easily reach the eastern edges of the Himalayas. Nevertheless, some contribution of maritime air masses coming from the Bay of Bengal is also visible in this region.

It appears that high d-excess values of daily rainfall collected along southern foothills of the Himalayas can be associated with air masses of very different origin. However, the common feature of almost all such cases is relatively long interaction of air masses with the continental surface, providing a chance to accommodate enough moisture of continental origin, characterized by elevated d-excess values, which then are transferred to local rainfall.

## 7 Data availability

All the isotope data used in this manuscript can be requested from R.D. Deshpande at desh@prl.res.in. Backward trajectory modelling was done for all daily precipitation events analysed in the framework of this study. The modelling results as well as the data used to construct footprint maps are available on request from M. Galkowski (Michal.Galkowski@fis.agh.edu.pl).

## 8 Authors contribution

G. Jeelani drafted the manuscript with the inputs from R.D. Deshpande, M. Galkowski and K. Rozanski. All the authors reviewed the manuscript and interpreted the data. M. Galkowski conducted Hysplit modelling.

## 9 Competing interests

The authors declare that they have no conflict of interest

## 10 Acknowledgements

Part of the sampling and the isotope analyses discussed in this study was undertaken under the aegis of IWIN National Programme (Deshpande and Gupta, 2008) funded jointly by the Department of Science and Technology (DST), Govt. of India, vide Grant No. IR/ S4/ESF-05/2004 and the Physical Research Laboratory (PRL). Authors acknowledge support of DST and PRL with gratitude. The IMD and CRIDA collected the rainwater samples from some of the stations included in this study. Dr. S.K. Peshin of IMD and Dr. V.U.M Rao of CRIDA are sincerely thanked for their coordination in sampling schedule. M. Galkowski and K. Rozanski were supported by the statutory funds of the AGH University of Science and Technology (project no. 11.11.220.01).

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




**Table 1**. General characteristics of the stations collecting daily precipitation samples for isotope analyses.

| Station code | Station name | Latitude/ Longitude | Altitude (m a.s.l.) | Mean annual temperature ($^{o}$C) | Mean annual precipitation (mm) | Sampling period | Number of samples |
|---|---|---|---|---|---|---|---|
| SGR | Srinagar | 34$^{°}$04´59"N/ 74$^{o}$47'50"E | 1595 | 13.6 | 693 | 04-2010/ 09-2011 | 121 |
| JMU | Jammu | 32$^{o}$39'22"N/ 74$^{o}$48'04"E | 267 | 24.2 | 1238 | 07-2009/ 06-2011 | 98 |
| PMR | Palampur | 32$^{o}$06'01"N/ 76$^{o}$32'49"E | 1275 | 19.1 | 2493 | 09-2008/ 09-2010 | 31 |
| RNC | Ranichauri | 30$^{o}$18'50"N/ 78$^{o}$24'25"E | 1872 | 15.1 | 1272 | 02-2009/ 12-2010 | 31 |
| JRH | Jorhat | 26$^{o}$43'21"N/ 94$^{o}$11'44"E | 99 | 24.0 | 2324 | 02-2010/ 12-2011 | 242 |
| DBR | Dibrugarh | 27$^{o}$29'05"N/ 95$^{o}$01'18"E | 111 | 23.2 | 2781 | 06-2009/ 10-2010 | 25 |





**Table 2.** Long-term (1985-2014) characteristics of surface air temperature and precipitation for Indian Summer Monsoon (ISM) and Western Disturbances (WD) periods, for the stations collecting daily precipitation samples for isotope analyses. Source of data: Srinagar, Jammu, Palampur, Jorhat, Dibrugarh - pl.climate-data.org; Ranichauri - Upadhyay et al., (2015).

| Station | Duration of ISM | $A_T$*) (°C) | ISM period | | WD period | | R**) |
|---------|----------|------------|--------|--------|--------|--------|------|
| | | | T (°C) | P (mm) | T (°C) | P (mm) | |
| Srinagar | July-Sept. | 23.1 | 23.0 | 150 | 10.5 | 543 | 0.28 |
| Jammu | July-Sept. | 21.0 | 29.7 | 854 | 22.4 | 384 | 2.22 |
| Palampur | July-Sept. | 17.2 | 23.2 | 1772 | 15.6 | 721 | 2.46 |
| Ranichauri | June-Sept. | 14.4 | 20.5 | 842 | 12.4 | 431 | 1.95 |
| Jorhat | May-Sept. | 12.3 | 28.1 | 1759 | 21.0 | 565 | 3.11 |
| Dibrugarh | May-Sept. | 11.7 | 27.1 | 2151 | 20.5 | 630 | 3.41 |

*) - peak-to-peak amplitude of long-term (1985-2014) seasonal changes of monthly surface air temperature at the station
**) - the ratio of cumulative rainfall amount collected at the given station during ISM and DW periods.





**Table 3.** Mean isotope characteristics of daily precipitation collected at six stations along southern foothills of the Himalayas and their relations to surface air temperature ($\Delta\delta^{18}O/\Delta T$) and precipitation amount (($\Delta\delta^{18}O/\Delta P$), for Indian Summer Monsoon (ISM) and Western Disturbances (WD) periods.

| Station | Period* | $\delta^{18}O$ ( ‰ ) | $\delta^{2}H$ ( ‰ ) | d-excess ( ‰ ) | N** | ($\Delta\delta^{18}O/\Delta T$)*** | ($\Delta\delta^{18}O/\Delta P$)*** |
|---|---|---|---|---|---|---|---|
| Srinagar | ISM | -6.55±0.99 | -43.8±7.6 | 8.6±0.9 | 23 | + | + – |
| | WD | -4.45±0.47 | -18.4±3.8 | 17.2±0.6 | 87 | + – | – |
| Jammu | ISM | -5.54±0.67 | -32.1±5.0 | 12.3±1.1 | 48 | + | + – |
| | WD | 0.05±0.68 | 10.2±4.8 | 9.8±1.1 | 31 | + | + – |
| Palampur | ISM | -8.70±1.25 | -60.9±10.5 | 8.7±0.9 | 19 | + – | + – |
| | WD | -1.39±0.42 | 0.2±3.2 | 11.3±1.4 | 12 | + – | + – |
| Ranichauri | ISM | -10.45±1.40 | -72.0±10.9 | 11.6±0.8 | 19 | + – | + – |
| | WD | -3.82±1.28 | -17.2±9.7 | 13.3±1.5 | 12 | + | + – |
| Jorhat | ISM | -1.97±0.27 | -43.5±2.8 | 11.5±0.3 | 174 | + – | + – |
| | WD | -6.88±0.34 | -1.2±2.3 | 14.6±0.5 | 68 | – | + – |
| Dibrugarh | ISM | -6.33±0.93 | -37.5±7.8 | 13.2±0.7 | 16 | – | n.d. |
| | WD | -3.01±1.54 | -11.2±12.4 | 13.1±0.6 | 9 | + – | n.d. |

\*) - duration of Indian Summer Monsoon (ISM) for Srinagar, Jammu and Palampur: July-September; for Ranichauri: June-September; for Jorhat and Dibrugarh: May-September. Rest of the year defined as Western Disturbances (WD).

\*\*) – number of daily precipitation samples analysed

\*\*\*) - ' + ' sign signifies positive, significant (R ≥ 0.4) correlation between $\delta^{18}O$ and daily temperature or precipitation
15 amount; ' – ' sign signifies negative, significant (R ≥ 0.4) correlation between $\delta^{18}O$ and daily temperature or precipitation amount; ' + – ' sign signifies lack of significant (R ≥ 0.4) correlation between $\delta^{18}O$ and daily temperature or precipitation amount.

n.d. - not determined



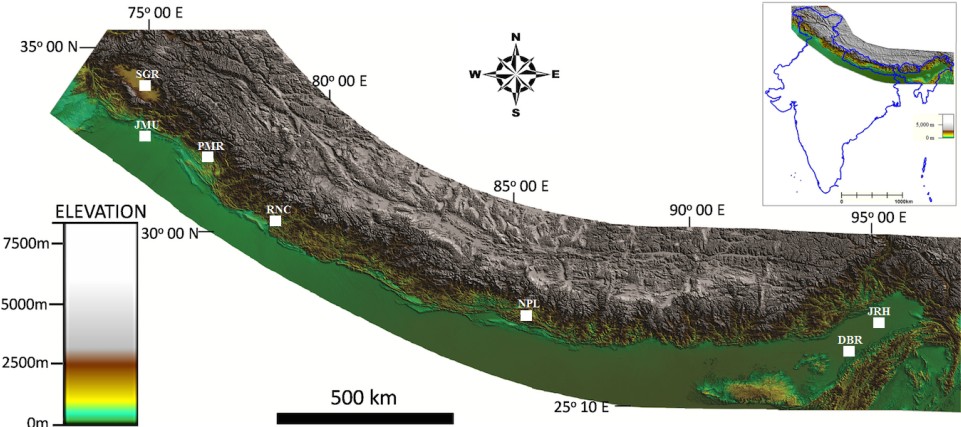

15 **Figure 1.** Locations of six sampling sites across the southern foothills of the Himalayas: SGR - Srinagar, JMU - Jammu, PMR - Palampur, RNC - Ranichauri, JRH - Jorhat and DBR - Dibrugarh. The position of Kathmandu station, Nepal (NPL), discussed in the text, is also marked in the figure.





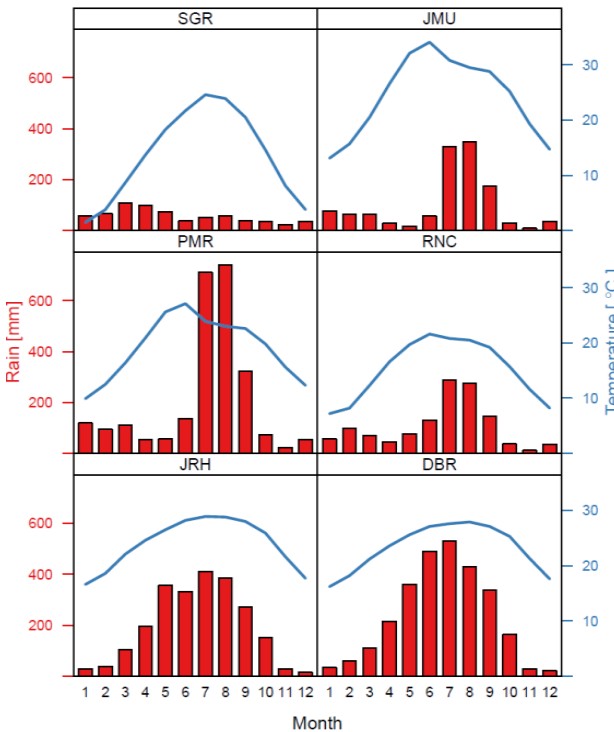

**Figure 2.** Long-term (1985-2014) monthly surface air temperature and precipitation data for six stations where daily
sampling of rainfall for isotope analyses was conducted. SGR - Srinagar, JMU - Jammu, PMR - Palampur, RNC -
10  Ranichauri, JRH - Jorhat, DBR - Dibrugarh. Source of data: Srinagar, Jammu, Palampur, Jorhat, Dibrugarh - pl.climate-
data.org; Ranichauri - Upadhyay et al., 2015.





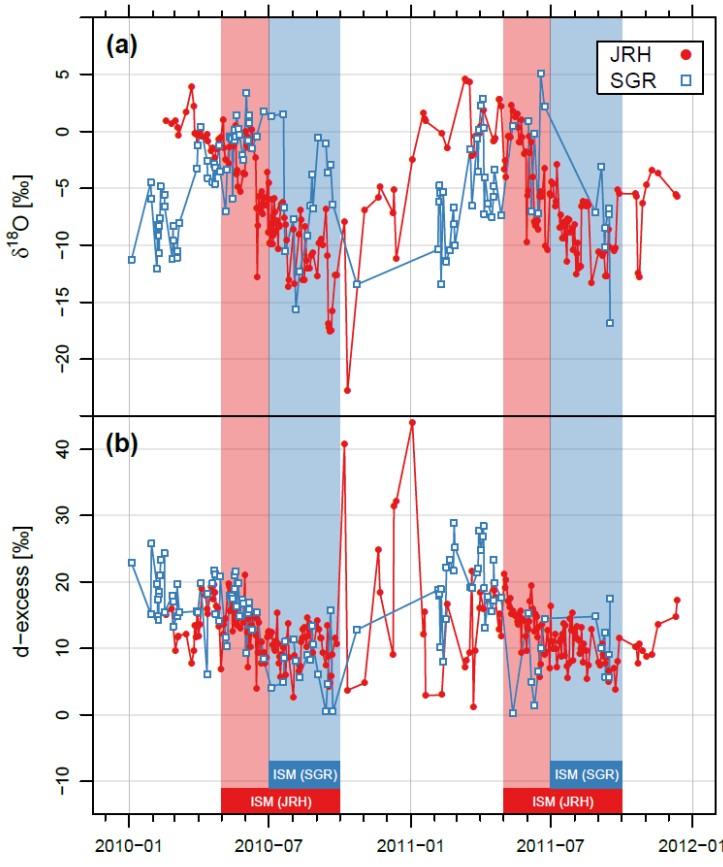

**Figure 3.** Seasonal variations of δ¹⁸O (a) and d-excess (b) of daily rainfall collected at Jorhat (JRH) and Srinagar (SGR)

5    stations. Indian Summer Monsoon (ISM) period at Srinagar (July-Sep) and Jorhat (May-Sep) is marked in blue and red

shading, respectively.



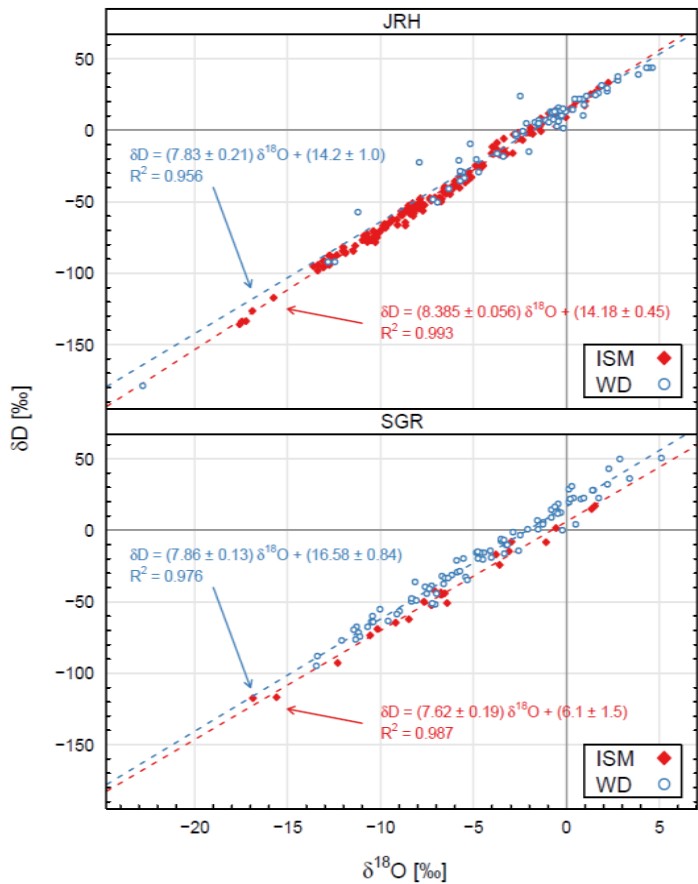

**Figure 4.** $\delta^2$H - $\delta^{18}$O relationship for daily isotope data available for Jorhat (JRH) and Srinagar (SGR) stations. Local
5  Meteoric Water Lines were calculated separately for Indian Summer Monsoon (ISM) and Western Disturbances (WD)
periods.



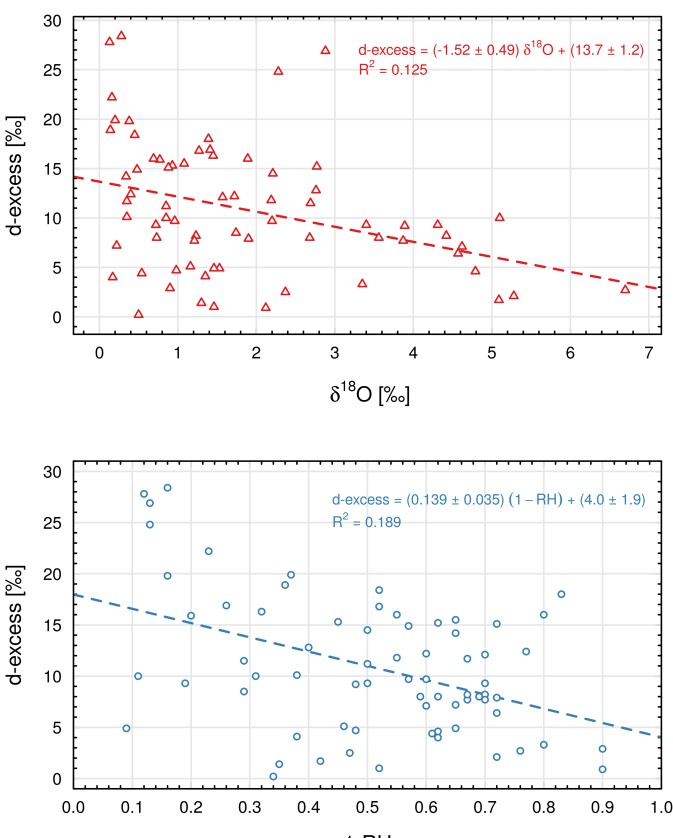

**Figure 5**. Upper diagram: relationship between deuterium excess and positive $\delta^{18}O$ values measured in daily precipitation samples collected by six stations distributed along southern foothills of the Himalayas (cf. Fig. 1). Lower diagram: relationship between deuterium excess and humidity deficit (1-RH, where RH stands for relative humidity) in the local atmosphere during precipitation events exhibiting positive $\delta^{18}O$ values.





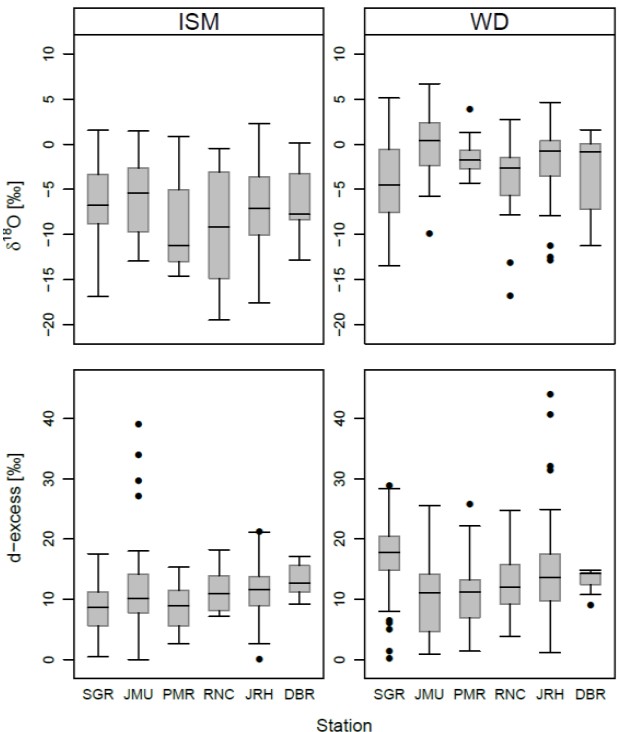

5  **Figure 6.** Box- and-whisker plots of $\delta^{18}O$ and d-excess values of daily rainfall at six stations (Srinagar - SGR; Jammu -JMU; Palampur - PMR; Ranichauri - RNC; Jorhat - JRH; Dibrugarth - DBR) collecting daily rainfall samples along southern foothills of the Himalayas (cf. Fig.1). The data are grouped into Indian Summer Monsoon (ISM) and Western Disturbances (WD) periods.





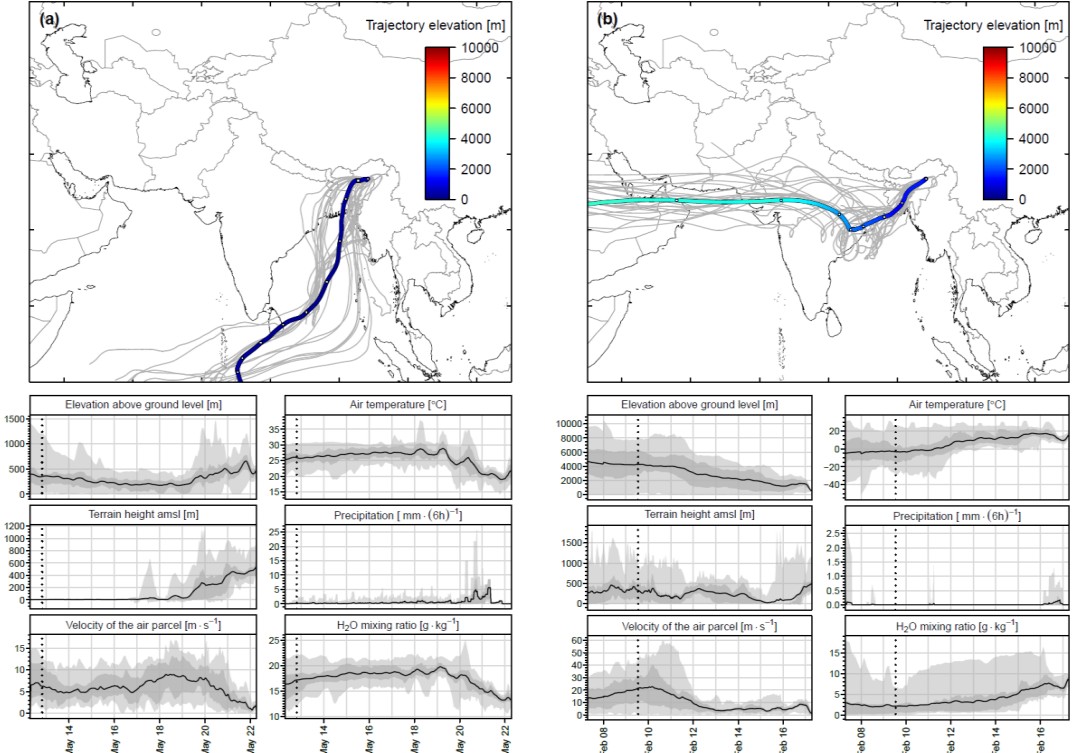

**Figure 7.** (a) upper diagram: 10-day backward trajectories arriving at Jorhat station on May 22, 2010, 12:00 hours and representing Indian Summer Monsoon (ISM) period. (b) upper diagram: 10-day backward trajectories arriving at Jorhat station on February 17, 2011, 12:00 hours and representing Western Disturbaces (WD) period. Twenty seven ensembles (grey lines in the background) and the mean trajectory (heavy line) are shown. Colours of the mean trajectory indicate elevation above the ground level in meters. Empty white dots on the mean trajectory indicate 24-hour intervals. Lower diagrams of (a) and (b) show the evolution of six selected parameters of air parcel along the trajectory. Evolution of the mean value of each parameter is marked in heavy black line and the associated uncertainty in grey. Dashed vertical lines mark the part of mean 10-day trajectory visible in the upper diagram.



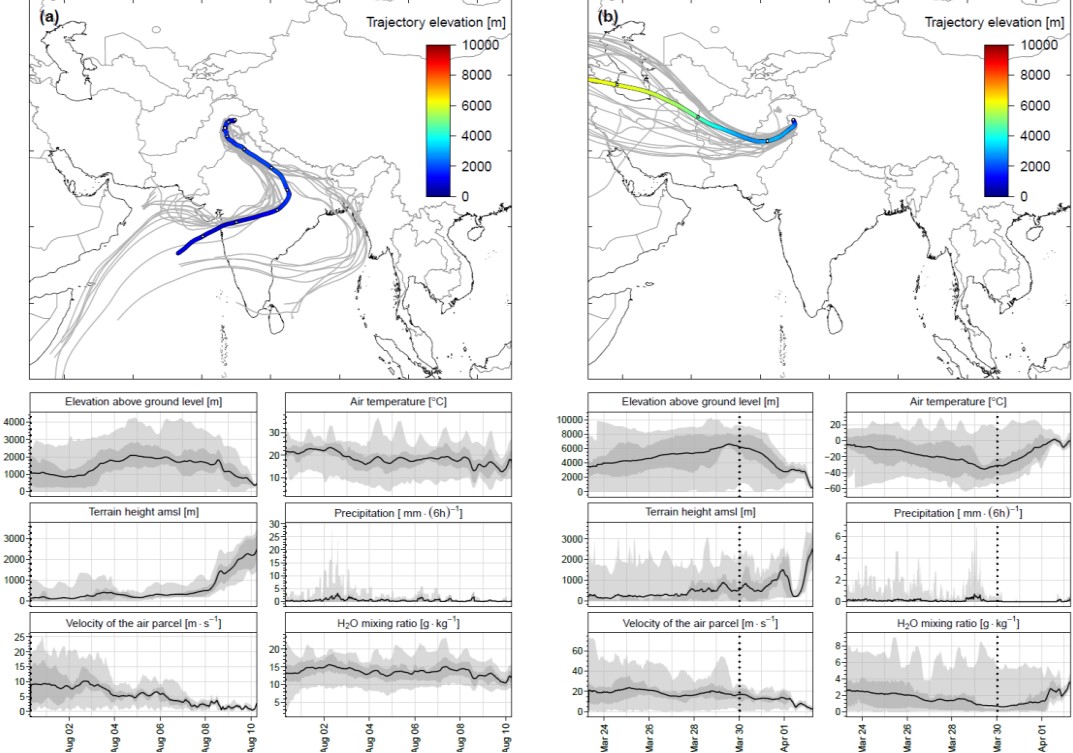

**Figure 8.** (a) upper diagram: 10-day backward trajectories arriving at Srinagar station on August 10, 2010, 12:00 hours and representing Indian Summer Monsoon (ISM) period. (b) Upper diagram: 10-day backward trajectories arriving at Srinagar station on April 2, 2011, 12:00 hours and representing Western Disturbaces (WD) period. Twenty seven ensembles (grey lines in the background) and the mean trajectory (heavy line) are shown. Colours of the mean trajectory indicate elevation above the local ground in meters. Empty white dots on the mean trajectory indicate 24-hour intervals. Lower diagrams of (a) and (b) show the evolution of six selected parameters of air parcel along the trajectory Evolution of the mean value of each parameter is marked by heavy black line and the associated uncertainty in grey. Dashed vertical lines mark the part of mean 10-day trajectory visible in the upper diagram.





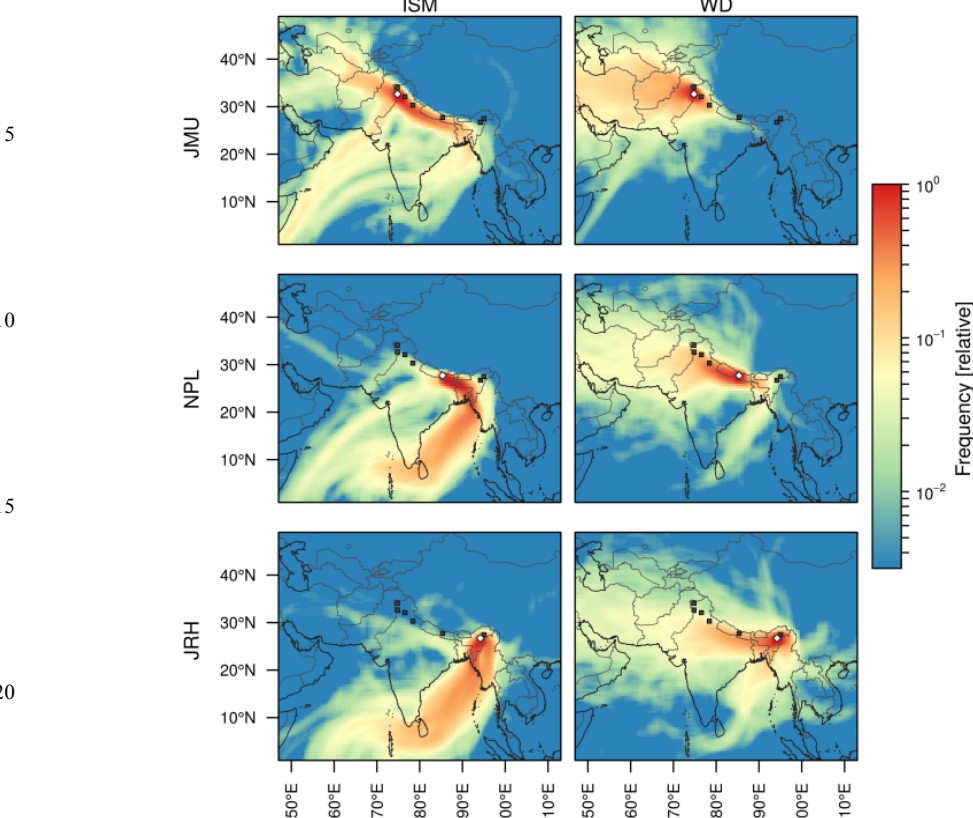

**Figure 9.** Footprint maps of air masses arriving at three stations: Jammu (JMU), Kathmandu (NPL) and Jorhat (JRH) representing western, central and eastern part of the studied transect, respectively (top, middle and bottom graphs). Separate maps were prepared for Indian Summer Monsoon (ISM) and Western Disturbances (WD) periods. Daily trajectories for the period 2009-2011 were reconstructed using Hysplit modelling framework (see text for details). International boundaries are only indicative and as provided by the software. Colour scale indicates the focally averaged number of trajectories passing through a grid cell.