# Peer review of "Isotopic composition of daily precipitation along southern foothills of the Himalayas: impact of marine and continental sources of atmospheric moisture"

_Atmospheric Chemistry and Physics, 2017_

## Referee Comment (RC1) · Anonymous Referee #1 · 6 Nov 2017

Evaluation of Manuscript

General comments: This MS presented isotopic composition of precipitation in the Himalaya region. The isotope data were analyzed using a backward trajectory analyses. The authors suggest that ISM evolution results in gradual decrease in isotope value, while WD period generally shows gradual increase in isotope value. The sampling locations are quite unique and important. Although this MS describe the data in detail, interpretation of the data is qualitative and descriptive. It is not clear the motivation of this MS and/or what is new and interesting in term of Atmospheric chemistry and

physics. In addition, the data selection criteria must be justified more clearly (see below). I feel that the content is more suitable for hydrological or meteorological journals rather than ACP.

Specific comments

P 4, L5-8, A station, Srinagar, shows almost no rain during the ISM season (JYly-September) (Figure 2).

There is no reason to assign the Srinagar station as a ISM affected station. Backward trajectory and isotope data at this station (in L11-13) should not be used as a reason because they are the data of this study which will be shown in Section 4.

P5, L6-9, "negative d-excess value were not considered in subsequent evaluation....in total, 35 isotope records were marked as locally affected....for Jammu station (19 out of 98)"

This is the most fundamental problem in this paper. How can you prove that there is no negative d-excess value in this region? Minima of d-excess values in Figure 3(b), Figure 5, and Figure 6 are zero. This is very unnatural. There should be many negative value data of d-excess, which did not show in the MS. The authors should justify this data selection criteria. For example, an data set of African monsoon event shows negative d-excess data (e.g., Risi et al., 2008) In addition, I suggest that all the data should be published as Supplementary data.

P5, L17-19, "ensemble members released at 12:00 LT on the days with precipitation sample collection"

The backward trajectory may change significantly before and during precipitation events. Thus, the fixed release time may cause some bias.

Figure 5 (bottom), Why you plotted (1-RH) not simple (RH)? Then, the dxs-RH regression line can be compared and discussed with the similar secondary evaporation effect found in African Monsoon region (Landais et al., 2010).

Technical corrections

P7, L1-20, These paragraphs appear to be simple description of the result. In fact, Fig3-5 were already described in Result section (4.1.). I feel that other paragraphs in discussion section are somewhat lengthy.

Table 1. "The number of samples" differs significantly in each station. I guess number of rain event differs. Thus, please add "number of precipitation day".

References

Risi, C., S. Bony, F. Vimeux, L. Descroix, B. Ibrahim, E. Lebreton, I. Mamadou, and B. Sultan (2008), What controls the isotopic composition of the African monsoon precipitation? Insights from event-based precipitation collected during the 2006 AMMA field campaign, Geophys. Res. Lett., 35, L24808, doi:10.1029/2008GL035920.

Landais et al., Combined measurements of 17Oexcess and d-excess in African monsoon precipitation: Implications for evaluating convective parameterizations, Earth and Planetary Science Letters, Volume 298, Issues 1–2, 15 September 2010, Pages 104-112, https://doi.org/10.1016/j.epsl.2010.07.033

---

## Author Comment (AC1) · 5 Jan 2018

Reply to interactive comments by Anonymous Referee #1

We appreciate very much the comments of Anonymous Referee #1. They allowed us to improve the overall shape of the MS and forced us to formulate our reasoning more clearly.

1. General comment As a general comment to the content of the MS, Referee #1 wrote: "This MS presented isotopic composition of precipitation in the Himalaya region.

[Figure]

The isotope data were analysed using a backward trajectory analyses. The authors suggest that ISM evolution results in gradual decrease in isotope value, while WD period generally shows gradual increase in isotope value. The sampling locations are quite unique and important. Although this MS describe the data in detail, interpretation of the data is qualitative and descriptive. It is not clear the motivation of this MS and/or what is new and interesting in term of Atmospheric chemistry and physics. In addition, the data selection criteria must be justified more clearly (see below). I feel that the content is more suitable for hydrological or meteorological journals rather than ACP".

Reply: Referee #1 suggests that interpretation of the data is qualitative and descriptive. The way of discussing isotopic composition of individual rainfall events from six stations extending over the distance of almost 3000 km and covering 3-year period (548 isotope data in total), presented in the MS, is in our opinion the only feasible way of gaining deeper insight into mechanisms and parameters controlling isotopic composition of precipitation in the important region of southern foothills of the Himalayas. Fully fledged modelling of isotope cycles in the regional atmosphere of Himalayas, with daily resolution and well represented continental feedback, would require vast amount of data not available to date and is beyond the reach of current crop of global general circulation models. Instead, the art of discussion presented in the MS and supported by extensive backward trajectory modelling provides in our view valuable, general understanding of physical factors controlling isotopic composition of precipitation in the Himalaya region. As to the appropriateness of publishing the MS in ACP we refer to the webpage of the ACP (Aims and Scope section, https://www.atmospheric-chemistry-and-physics.net/about/aims_and_scope.html) where subject 'precipitation' and 'isotopes' is explicitly mentioned: "The main subject areas comprise atmospheric modelling, field measurements, remote sensing, and laboratory studies of gases, aerosols, clouds and precipitation, isotopes, radiation, dynamics, biosphere interactions, and hydrosphere interactions (for details see journal subject areas). The journal scope is focused on studies with general implications for atmospheric science rather than investigations that are primarily of local or technical interest". Therefore, we leave the question of

appropriateness of the MS to the judgement of the editors of the ACP.

2. Specific comments

P 4, L5-8 "A station, Srinagar, shows almost no rain during the ISM season (July-September) (Figure 2). There is no reason to assign the Srinagar station as a ISM affected station. Backward trajectory and isotope data at this station (in L11-13) should not be used as a reason because they are the data of this study which will be shown in Section 4."

Reply: In the revised MS we have made reference to Section 4.1 (Fig.3 and Table 3) and Section 5.1 (Fig. 7) to support the statement. In fact, in our view, the characteristic decline of $\delta$18O ($\delta$2H) values at the onset of ISM period (cf. Srinagar $\delta$18O record in Fig. 3) is a powerful tool to identify arrival of monsoonal air masses even in situations when precipitation amount does not indicate this. We underlined this fact in the conclusions.

P5, L6-9 "negative d-excess value were not considered in subsequent evaluation: : :.in total, 35 isotope records were marked as locally affected: : :.for Jammu station (19 out of 98)" This is the most fundamental problem in this paper. How can you prove that there is no negative d-excess value in this region? Minima of d-excess values in Figure 3(b), Figure 5, and Figure 6 are zero. This is very unnatural. There should be many negative value data of d-excess, which did not show in the MS. The authors should justify this data selection criteria. For example, an data set of African monsoon event shows negative d-excess data (e.g., Risi et al., 2008) In addition, I suggest that all the data should be published as Supplementary data"

Reply: Isotopic composition of maritime water vapour is formed in the process of evaporation of surface ocean water. This process is well described by Craig and Gordon model (see for instance review paper by Horita et al. 2007) where two fractionations steps are postulated: (i) equilibrium fractionation between liquid and vapour phase of water, and (ii) kinetic fractionation step, which is linked with transport of water vapour

through laminar layer of the atmosphere adjacent to the interface. Whereas the ratio of equilibrium fractionations for 2H and 18O isotopes is approximately 8 and causes the equilibrium vapour to be located in the $\delta$2H - $\delta$18O space on the line with the slope 8 and intercept zero, the kinetic fractionation step which is characterized by the ratio of kinetic 2H and 18O fractionations equal ca. 0.9, moves this vapour to the line with the slope equal 8 and intercept equal 10. Since subsequent in-cloud condensation is considered to be equilibrium process, the rainfall, which is leaving the cloud base will stay on this line with the slope of 8. This is why the Global Meteoric Water Line is described by the equation $\delta$2H = 8âŃĚ$\delta$18O +10 (see for instance seminal paper by Merlivat and Jouzel, 1979). If rainfall is formed from continental moisture sources under transient conditions, the isotope mass balance considerations suggest that it should have generally higher d-excess values than the original water because vapour generated in such processes will have d-excess values higher than 10. From the above discussion it is clear that it would be virtually impossible to generate negative d-excess values in typical regional rain-forming processes. As we were interested in such processes, we decided to remove from the discussion all the data for daily rainfall events exhibiting negative d-excess values. We did not said in the MS that "there is no negative d-excess value in the region". We only stated that those are most probably locally influenced data (below cloud base evaporation of rain drops and/or evaporation of collected rainfall in the rain gauge). As our focus in the MS was clearly on regional processes, such decision was in our view justified. In the section 7 (Data availability) we will mention that all data, including those, which were not considered in the discussion, will be available on request.

P5, L17-19, "ensemble members released at 12:00 LT on the days with precipitation sample collection" The backward trajectory may change significantly before and during precipitation events. Thus, the fixed release time may cause some bias.

Reply: Good point. Unfortunately, we had no information about exact timing of precipitation events (except of the date). Thus, we were left with two options. One was

to calculate a single release starting on a given hour (midday was selected), which has the potential problem of representativeness that Referee #1 mentioned. The other option was to calculate several releases during the selected day and use them all in the analyses. For footprint analyses, the timing of release has little significance, as we used 3 years of daily releases. We wanted to represent large scale transport patterns here, and there is no reason to expect that night-time releases would show a different pattern. In case of individual rainfall events, adding additional trajectory ensembles would be most meaningful in case of frontal passages. However, one has to remember the resolution of meteorological data driving Hysplit analysis (3h temporal, 1deg x 1deg spatial), which already added uncertainty to the transport analysis. We have considered and estimated this uncertainty by using the ensemble analysis. Even if a strong frontal system would be passing over the station on a given day, it is likely that the ensemble scheme would capture at least part of the variability of the transport patterns. In fact we took a closer look at the events presented in Fig.7 and Fig.8, generating the trajectories for the whole given day (every three hours). The results did not display significant changes in ensemble patterns – neither in terms of trajectory source areas, nor in the behaviour of presented meteorological variables. In the revised manuscript we will include the paragraph discussing the question of representativeness of backward trajectory analysis for individual rainfall events.

Figure 5 (bottom) " Why you plotted (1-RH) not simple (RH)? Then, the dxs-RH regression line can be compared and discussed with the similar secondary evaporation effect found in African Monsoon region (Landais et al., 2010)"Discussion paper

Reply: In the framework of Craig-Gordon evaporation model the actual magnitude of kinetic fractionation is controlled by humidity deficit (1-RH), not by RH. Therefore, we considered it more appropriate to relate the d-excess of individual rainfall events to 1-RH. In this representation the d-excess values tend to drop with increasing humidity deficit, as one would expect it for partial evaporation of raindrops.

3. Technical corrections

P7, L1-20, "These paragraphs appear to be simple description of the result. In fact, Fig. 3-5 were already described in Result section (4.1.). I feel that other paragraphs in discussion section are somewhat lengthy"

Reply: We tried to be rigorous and separate the presentation of the generated data in the form of Tables and Figures from the discussion of those. We feel that this way makes the text and the overall appearance of the MS more transparent and digestible. In the Discussion section we tried to be as comprehensive as possible, and explain all aspects of the presented data in sufficient detail. In our view, significant shortening of the Discussion section would have a negative impact on the overall value of the MS.

Table 1. "The number of samples" differs significantly in each station. I guess number of rain event differs. Thus, please add "number of precipitation day".

Reply: Labelling of last column in Table 1 will be modified accordingly.

---

## Referee Comment (RC2) · M. Schulz (Referee) · 19 Feb 2018

The paper presents a valuable dataset on water isotope composition in the Himalayas. The subject is probably at the edge of ACP interests, but the inner tracer physics of monsoon precipitation should be of interest to AC readers. Unfortunately it was difficult to have a second reviewer, so the editor jumps in here, and recommends major revision. See my detailed comments below.

In accordance with reviewer 1 I would suggest to document more of the data: Figure 3

[Figure]

type of plots should be shown for all stations, at least as supplementary figures. The data corresponding to negative d excess data should be shown as well and be included in the analysis.

"The available isotope dataset was screened for negative values of the d-excess." => All negative d-excess data should be included in the analysis, eg by presenting two workups, one with and one without the negative d-excess values. There is no reason to set the threshold for excluding data to zero deuterium excess.

"A secondary isotope parameter, deuterium excess (d = $\delta$2H - 8âŃĚ$\delta$18O; Dansgaard, 1964) defines the position of data points in the $\delta$2H-$\delta$18O space with respect to GMWL". => The offset of 10 is in the GMWL definition, but not in the deuterium excess definition. This seems not consistent to me. Maybe rephrase to "with respect to the slope of the GMWL using no offset".

Figure 5 is not very convincing as a regression with low correlation coefficients, and I doubt it explains much. One could maybe bin the data into two classes to get a more meaningful characterisation of the variability.

figure 5 The unit is also not totally clear (1-RH) , equals 0.9 10% R.H.? And furthermore: Does it rain in such dry conditions? Is it the RH on the day of the precipitation?

"the linear relationship between $\delta$2H and $\delta$18O is generally better defined pointing to moisture sources of similar nature and similar conditions of rainfall formation"=> Not clear - there are not different 'moisture' sources for $\delta$2H and $\delta$18O, clarify please.

"to gradual reduction of 2H and 18O content in the marine moisture"=> the word marine here is confusing. Its the transported moisture which is a mix of isotope depleted moisture and fresh marine moisture, or?

"Jammu. This station is located at western edge of the transect, far away from oceanic sources of moisture." => I dont think the other two stations nearby (Palampur and Ranichauri) are closer to oceanic sources. Please clarify.

"The fact that d-excess values are inversely correlated with $\delta$18O values and decrease rising humidity deficit of the local atmosphere (Fig. 5)," => please rephrase sentence

Discussion on high d-excess values => The mechanismn for getting to high d-excess values is not clear to me. Should be better explained.

"Superimposed on this general trend are short-term fluctuations of the isotopic composition of rainfall having their roots in local effects." => I think the local effects need to be better described, defined. What is a local effect?? actually throughout the text.

"These peculiar isotope characteristics can be explained only when dominating continental origin of moisture is postulated. " => not clear why continental origin per se means enriched isotopes.

"Water stored in the soil during ISM period is returned to the local atmosphere during WD period through evapotranspiration processes." => yes, but why is this mentioned here?

---

## Author Response (AR1)

**Reply to interactive comments on "Isotopic composition of daily precipitation along southern foothills of the Himalayas: impact of marine and continental sources of atmospheric moisture" by Ghulam Jeelani et al.**

**Reply to interactive comments by M. Schulz**

Comments by Co-Editor of ACP, M. Schulz helped us to improve considerably the overall shape of the manuscript. We very much appreciate his time and efforts devoted to this task.

Specific comments

1. *In accordance with reviewer 1 I would suggest to document more of the data: Figure 3 type of plots should be shown for all stations, at least as supplementary figures. The data corresponding to negative d excess data should be shown as well and be included in the analysis. All negative d-excess data should be included in the analysis, eg by presenting two*
*workups, one with and one without the negative d-excess values. There is no reason*
*to set the threshold for excluding data to zero deuterium excess.*

Reply:
Although inclusion of negative d-excess values did not altered in any significant way our major findings and conclusions, we accept the criticism of both reviewers and now all available isotope data are presented and discussed in the revised MS.

2. *The offset of 10 is in the GMWL definition, but not in the deuterium excess definition. This seems not consistent to me. Maybe rephrase to "with respect to the slope of the GMWL using no offset.*

Reply:
There is no inconsistency in the definition of the d-excess proposed by Dansgaard in 1964.
The d-excess defines the position of the given data point in the $\delta^2 H$- $\delta^{18}O$ space with respect to the Global Meteoric Water Line defined by the equation $\delta^2 H = 8 \cdot \delta^{18}O + 10$. According to the definition of d-excess ($d = \delta^2 H - 8 \cdot \delta^{18}O$) all points forming GMWL have this parameter equal +10‰. Data points lying outside the line will have higher or lower values of this parameter. Deuterium excess is often mixed-up with the intercept of the global (or local) meteoric water lines. Local meteoric water line may have the intercept equal +10 but this does not mean that points forming this line will have the d-excess values also equal +10.

3. *Figure 5 is not very convincing as a regression with low correlation coefficients, and I*
*doubt it explains much. One could maybe bin the data into two classes to get a more*
*meaningful characterisation of the variability.*
*figure 5 The unit is also not totally clear (1-RH) , equals 0.9 10% R.H.? And furthermore:*
*Does it rain in such dry conditions? Is it the RH on the day of the precipitation?*

Reply:
Fig. 5 is retained in the revised manuscript (now Fig. 8). Despite of low correlation coefficients characterizing linear fits of the data presented in Fig. 5 we consider those data important for our discussion. To make the comparison with West African Monsoon data (as suggested by Referee #1) more straightforward, we switched to relative humidity expressed in %. In the revised MS we discuss in detail d-excess/relative humidity relationships obtained for both monsoon systems.

4. "*The linear relationship between $\delta^2H$ and $\delta^{18}O$ is generally better defined pointing to moisture sources of similar nature and similar conditions of rainfall formation"=> Not clear - there are not different 'moisture' sources for $\delta^2H$ and $\delta^{18}O$, clarify please.*

Reply:
The revised text has been rephrased accordingly.

5.*"to gradual reduction of $^2H$ and $^{18}O$ content in the marine moisture"=> the word marine here is confusing. Its the transported moisture which is a mix of isotope depleted moisture and fresh marine moisture, or?*

Reply:
The revised text has been rephrased accordingly.

6. *"Jammu. This station is located at western edge of the transect, far away from oceanic sources of moisture." => I dont think the other two stations nearby (Palampur and Ranichauri) are closer to oceanic sources. Please clarify.*

Reply:
The revised text has been rephrased accordingly.

7. *"The fact that d-excess values are inversely correlated with $\delta^{18}O$ values and decrease rising humidity deficit of the local atmosphere (Fig. 5)," => please rephrase sentence*

Reply:
The revised text has been rephrased accordingly

8. *Discussion on high d-excess values => The mechanisms for getting to high d-excess values is not clear to me. Should be better explained.*

Reply:
The text has been modified accordingly to make it more clear. It would be difficult and is also not necessary to lay out here all the details related to description of isotope effects accompanying evaporation process of water. They are well established and described in the relevant literature, some of which is quoted in the revised MS.

9. *"Superimposed on this general trend are short-term fluctuations of the isotopic composition of rainfall having their roots in local effects." => I think the local effects need to be better described, defined. What is a local effect?? actually throughout the text.*

Reply:
The revised text has  been revised accordingly to clarify this issue.

10. *"These peculiar isotope characteristics can be explained only when dominating continental origin of moisture is postulated. " => not clear why continental origin per se means enriched isotopes*.

Reply:
This issue has been clarified throughout the revised text

11. *"Water stored in the soil during ISM period is returned to the local atmosphere during*
*WD period through evapotranspiration processes." => yes, but why is this mentioned*
*here?*

5   *Reply:*
It is mentioned here because these processes generate vapour, which is significantly enriched in heavy isotopes when compared to unaltered vapour of oceanic origin (cf. two lines below the questioned phrase in the original text). The revised text has been modified and extended to make it more clear.

[revised manuscript text omitted]

---

## Author Response (AR2)

**Reply to final suggestion by Prof. M Schulz on "Isotopic composition of daily precipitation along southern foothills of the Himalayas: impact of marine and continental sources of atmospheric moisture" by Ghulam Jeelani et al.**

**Reply to comment by M. Schulz**

We very much appreciate his time and efforts devoted to this task.

1. *I would suggest to revise and split figure S1. It is not lisible as it is. You should remove the stations JRH and SGR, as these are already in figure 3 in the min manuscript. And then please split the figure in two, one covering stations DBR and JMU and another one PMR and RNC.*

Reply:
Thank you very much for the suggestion. We have removed the stations JRH and SGR from the figure S1 as these stations are already similarly plotted in figure 3. After removing the two stations the figure is quite lisible, so we thought not to further split it.